# The 4D Nucleome Data Portal as a resource for searching and visualizing curated nucleomics data

Sarah B. Reiff [1], Andrew J. Schroeder [1], Koray Kırlı[1], Andrea Cosolo [1], Clara Bakker [1], Luisa Mercado[1], Soohyun Lee[1], Alexander D. Veit[1], Alexander K. Balashov[1], Carl Vitzthum[1], William Ronchetti[1], Kent M. Pitman [1], Jeremy Johnson[1], Shannon R. Ehmsen[1], Peter Kerpedjiev[1], Nezar Abdennur[2], Maxim Imakaev [2], Serkan Utku Öztürk[3], Uğur Çamoğlu[3], Leonid A. Mirny [2,4], Nils Gehlenborg [1], Burak H. Alver [1] & Peter J. Park [1✉]

The 4D Nucleome (4DN) Network aims to elucidate the complex structure and organization of chromosomes in the nucleus and the impact of their disruption in disease biology. We present the 4DN Data Portal (https://data.4dnucleome.org/), a repository for datasets generated in the 4DN network and relevant external datasets. Datasets were generated with a wide range of experiments, including chromosome conformation capture assays such as Hi-C and other innovative sequencing and microscopy-based assays probing chromosome architecture. All together, the 4DN data portal hosts more than 1800 experiment sets and 36000 files. Results of sequencing-based assays from different laboratories are uniformly processed and quality-controlled. The portal interface allows easy browsing, filtering, and bulk downloads, and the integrated HiGlass genome browser allows interactive visualization and comparison of multiple datasets. The 4DN data portal represents a primary resource for chromosome contact and other nuclear architecture data for the scientific community.

[1] Department of Biomedical Informatics, Harvard Medical School, Boston, MA 02115, USA. [2] Institute for Medical Engineering and Science, Massachusetts Institute of Technology, Cambridge, MA 02139, USA. [3] Karya SMD Software Solutions, İzmir 35040, Turkey. [4] Department of Physics, Massachusetts Institute of Technology, Cambridge, MA 02139, USA. ✉email: peter_park@hms.harvard.edu

The 4D Nucleome (4DN) Network[1] is an NIH Common Fund project that started in 2015 with the overarching goal of elucidating the three-dimensional organization of chromosomes in the nucleus across different cell types and cell cycle stages, and to understand how perturbation of this structure can impact human health in cases of cancer and other diseases. The first phase (2015–2020) of the consortium included five projects on nucleomics technology development, nine projects on the development of imaging tools, six projects studying nuclear bodies and compartments, six centers studying nuclear organization, and 10 additional collaborating projects, involving ~100 laboratories in total. The second phase of the consortium (2020–2025) includes eight projects studying real-time chromatin dynamics, 16 projects examining the role of nuclear biology in human health and disease, and four centers developing methods for data integration, modeling, and visualization. The new projects bring the total of 4DN network laboratories to over 150 and promise the generation of even more nucleomics data, particularly with relevance to human health, as well as more advanced modes of data integration across disparate assay types.

The 4DN Data Coordination and Integration Center (DCIC) has created a web portal that serves as a repository for data generated by the 4DN Network members. It has also imported external datasets that are widely used by the community. Implemented fully on the Amazon Web Services (AWS) platform with the latest technologies, the portal has been engineered to provide utility to the broader community of nuclear biology researchers. It enables easy searching and browsing of the data and, importantly, the associated metadata, thus allowing for increased reproducibility both at the analytical and experimental levels.

The portal was designed to accommodate data generated from both genomics and microscopy experiments. A large portion of the genomics data comes from chromosome conformation capture (3C) assays such as Hi-C[2]. Additional data derived from several other genomic assay types that probe chromosome conformation or other aspects of nuclear structure and function, such as replication timing, chromatin accessibility, or spatial proximity of DNA to other cellular components. These assays include Repli-seq[3], DamID[4], CUT&RUN[5], and SPRITE[6]. We have incorporated elements specific to imaging modalities into our data model as well, and provide visualization capabilities for some imaging data types. Many current datasets use imaging as a tool for measuring distances between genomic locations or between a genomic region and a sub-nuclear structure, and imaging is often used as a cross-validation technique for the results of genomic assays. Current imaging data types available include standard fluorescent in situ hybridization (FISH) that targets a DNA or RNA sequence, multiplexed FISH that can target many loci in a single fixed sample[7], and dynamic single-particle tracking.

In addition to making data easily accessible to users, the 4DN DCIC has aimed to develop and run standardized bioinformatics pipelines on submitted raw data to generate consistent and comparable results. These can then be explored in an integrated HiGlass browser that supports visualization of 2D contact maps and linear 1D tracks such as gene annotations or CUT&RUN peaks[8]. The portal also supports the download of raw and processed files for users that want to use the files locally.

Here we describe the 4DN Data Portal, a repository for genomic and microscopy nuclear architecture datasets. We discuss data that can be found in the portal, interactive visualization capabilities for these datasets in the portal, and utility to users outside the 4DN network.

## Results

**4DN Data at the data portal**. The majority of datasets in the portal are submitted by the 4DN researchers who performed the studies. A subset of datasets on the 4DN data portal was generated outside the consortium but was curated by 4DN curators because they were widely used in the community; these data were uploaded from other public repositories such as NCBI's Gene Expression Omnibus (GEO) and Sequence Read Archive (SRA). External users interested in submitting their data or those wishing to suggest a dataset for inclusion in the portal may contact our support desk (support@4dnucleome.org).

Within the data model of the 4DN data portal, the top-level item is called an Experiment Set, and represents a set of replicate experiments, all performed under the same experimental conditions; as such, quantities of data will be discussed below in terms of experiment sets. 4DN data generators are strongly encouraged to include multiple biological replicates for every experiment, and currently, over 800 experiment sets in the portal have met this standard.

**Genomics datasets**. For genomics assays, Hi-C and its variants make up the largest proportion of experiments in the 4DN data portal. The original Hi-C protocol was reported in 2009[2], as a chromosome conformation capture (3C) method that assays pairwise interactions in DNA across the whole genome by making use of high-throughput sequencing. Since then, many variants of Hi-C have been described. The first Hi-C experiments used 1% formaldehyde cross-linking, a six-cutter restriction enzyme, and digestion and ligation steps were performed in lysed nuclei in a dilute solution. Later, an in situ version involved performing the digestion and ligation steps in intact nuclei. Other types of Hi-C assays involve using different cross-linking chemistries and chromatin fragmentation methods (Micro-C, DNase Hi-C, Hi-C 3.0)[9–11], and single-cell and single nucleus versions of Hi-C have also been implemented[12–14]. In total, the 4DN data portal has 610 public Hi-C experiment sets, spanning nine subtypes of Hi-C assays (Table 1). Some of these datasets are particularly useful because of their high sequencing depth. Over 40 Hi-C experiment sets on the portal have over 1 billion read pairs after filtering, and a table of these is available on the data portal resource pages (https://data.4dnucleome.org/resources/data-collections/high-resolution-hic-datasets).

The same year that Hi-C was reported, the first ChIA-PET (Chromatin Interaction Analysis by Paired-End Tag sequencing) experiments were published. Similar to Hi-C in design, ChIA-PET uses chromatin immunoprecipitation to enrich DNA-protein complexes and then employs proximity ligation and sequencing, enabling the discovery of chromatin interactions mediated by a target protein[15,16]. Together with ChIA-Drop, a droplet-based assay similar to ChIA-PET[17], and PLAC-seq, another similar assay in which proximity ligation is performed prior to immunoprecipitation[18], IP-based 3C assays comprise 34 public experiment sets on the 4DN data portal (Table 1).

Recently, a great deal of technology has been developed by members of the 4DN community, resulting in new innovative assays for probing aspects of nuclear DNA structure. To give a few examples, SPRITE is a method for identifying multi-way interactions between distal genomic regions[6]; MARGI is a method for mapping RNA-chromatin interactions[19–21]; CUT&RUN is a DNA-binding assay generating data similar to ChIP-seq but which produces much lower background[5]; and TSA-seq is being used to analyze the proximity of nuclear structures to genomic regions on a genome-wide scale[22]. All the genomic assays currently on the 4DN data portal and available to the public are shown in (Table 1).

**Table 1 Genomic assay types in the 4D Nucleome Data Portal.**

| Experiment type | No. of public experiment sets |
|---|---|
| **Hi-C** | **610** |
| in situ Hi-C | 335 |
| Dilution Hi-C | 118 |
| DNase Hi-C | 21 |
| Micro-C | 26 |
| Single-cell Hi-C | 11 |
| Single nucleus Hi-C | 17 |
| sci-Hi-C[13] | 28 |
| Capture Hi-C[53] | 40 |
| TCC | 14 |
| **IP-based 3C assays** | **34** |
| ChIA-PET | 4 |
| in situ ChIA-PET | 10 |
| ChIA-Drop[17] | 2 |
| PLAC-seq[18] | 18 |
| **DNA-binding assays** | **202** |
| ChIP-seq | 141 |
| CUT&RUN | 61 |
| CUT&Tag[54] | 2 |
| **Other sequencing assays** | **421** |
| Repli-seq[55] | 138 |
| SPRITE[6] | 3 |
| DamID | 66 |
| ATAC-seq | 21 |
| RNA-seq | 90 |
| TRIP[56] | 7 |
| NAD-seq[57] | 8 |
| TSA-seq[22] | 67 |
| MARGI[19] | 6 |
| GAM[58] | 6 |
| RE-seq(DpnII-seq[59]) | 11 |
| Bru-seq[60] | 1 |
| MC-3C[61] | 1 |
| **Total** | **1273** |

Bold rows indicate categories of genomic assays and their subtotal counts.

**Genomics pipelines and QC.** For genomics datasets, data are submitted as raw fastq files. All data are aligned to the genome references hg38 and mm10 for analyses of human and mouse samples, respectively. For Hi-C experiments, the general analysis pipeline involves (1) aligning reads to a reference genome; (2) filtering aligned reads; (3) combining replicates for a single list of contact pairs; and (4) matrix aggregation and normalization. At each of these steps, different tools and parameters are employed by independent laboratories. Although optimal tools/parameters may depend on the type of downstream analysis performed, a uniform processing pipeline was necessary to ensure consistency and compatibility. After extensive discussions and testing by the members of the 4DN Analysis Working Group, the final version of the Hi-C processing included the following steps: (1) alignment by BWA MEM[23] with the -SP5M option to ensure that paired reads are aligned independently but the results are formatted properly as paired-end data and that the 5' portion of a chimeric alignment is reported as a primary soft-clipped alignment, (2) sorting and filtering the reads using pairtools[24] and (3) aggregating filtered reads into a contact matrix and normalizing it. The analysis also includes quality control steps using Fastqc[25] and Pairsqc[26]. This pipeline outputs a multi-resolution contact matrix at the 4DN standard resolutions of 1 kb, 2 kb, 5 kb, 10 kb, 25 kb, 50 kb, 100 kb, 250 kb, 500 kb, 1 Mb, 2.5 Mb, 5 Mb and 10 Mb. This matrix is generated using the cooler software and the .mcool file format[27], which is compatible with the HiGlass interactive 2D

genome browser[8] and can be visualized on the portal. An additional contact matrix is also generated in .hic format, which is compatible with the Juicebox 2D interactive genome browser[28]. More details of the pipeline can be found in the portal's resource pages at https://data.4dnucleome.org/resources/data-analysis/hi_c-processing-pipeline.

Genomic contact matrices generally show evidence of genomic compartments[2] as well as local regions of enriched intra-compartment contacts, known as topologically-associated domains (TADs)[29,30]. There is a great deal of interest in exploring the nature and dynamics of TADs and sub-TADs, domains nested within others. Identification of such domains is difficult due to the lack of validated sets, and detection algorithms are continually being refined[31–33]. At this time, the 4DN Data Portal runs two domain identification workflows on contact matrices to report compartments and TAD boundaries, using the cooltools software[34]. The first workflow uses an eigenvector decomposition of the matrix to call active (A) and inactive (B) compartments. For cis contacts, eigenvalues are calculated by the cooltools command cooler_cis_eig with default values of n_eigs=3, phasing_track_col='GC', ignore_diags=None, clip_percentile=99.9, and sort_metric='var_explained'. For trans contacts, the cooler_trans_eig command is used with the default parameters n_eigs=3, partition=None, phasing_track_col='GC', and sort_metric='var_explained'. The second workflow computes insulation scores along the diagonal of the matrix, based on average interaction frequencies crossing over each genomic bin, and prominent dips in this score indicate boundaries between domains[35]. The insulation table is calculated with the cooltools calculate_insulation_score command with default parameter window=100000, followed by the find_boundaries command with default parameters pixels_frac=0.66 and cutoff=2. As the nature of TADs and domains is an area of active investigation, the 4DN DCIC chooses to report boundaries based on insulation scores to provide results that might be further utilized in developing domain-detection algorithms. Results of both workflows are accessible as bed files as well as bigwig files which can be visualized in HiGlass on the data portal. More details on the domain-detection pipelines can be found at https://data.4dnucleome.org/resources/data-analysis/insulation_compartment_scores.

Processing pipelines are run on other types of high-throughput sequencing assays at the portal as well, including Repli-seq, CUT&RUN, and MARGI (Table 2). The pipeline for Repli-seq involves (1) trimming reads with cutadapt and aligning to a reference genome with bwa-mem[23]; (2) filtering valid alignments with samtools[36]; and (3) binning and aggregating for 5kb windows with bedtools[37]. The final output is provided in gzipped bedgraph and bigWig formats.

The CUT&RUN pipeline on the data portal processes paired-end reads in three main steps: (1) reads are processed with Trimmomatic for quality filtering and adapter trimming, and aligned to a reference genome with bowtie2[38]; (2) duplicate reads are filtered out using Picard[39] and samtools[36], and converted into .bed format with bedtools[37]; and (3) peaks are called with SEACR[40]. The final outputs of the pipeline include a peaks file and a bigWig track. The final bigWig files of the CUT&RUN and Repli-seq pipelines can both be visualized on the portal as 1D tracks in the HiGlass browser. The MARGI pipeline is similar to the Hi-C pipeline, and is adapted from the original pipeline written by the iMARGI creators[21].

For ATAC-seq, ChIP-seq, and RNA-seq data, pipelines from the ENCODE Data Coordination Center[41] have been adapted for the 4DN platform. For ATAC-seq and ChIP-seq, the final output of these pipelines includes a bigwig file containing the fold change in signal across the genome, which can be

**Table 2 4D Nucleome analysis pipelines.**

| Pipeline | Steps | Software | Available file formats | CWL/WDL filename |
|---|---|---|---|---|
| Hi-C[1] | Alignment | bwa-mem | .bam | bwa-mem.cwl |
| | Filtering | pairtools | .pairs | hi-c-processing-bam.cwl |
| | Merging replicates & matrix aggregation | cooler | .hic, .mcool | hi-c-processing-pairs.cwl |
| MARGI[2] | Alignment | bwa-mem | .bam | imargi-processing-fastq.cwl |
| | Filtering | pairtools | .pairs | imargi-processing-bam.cwl |
| | Merging replicates & matrix aggregation | cooler | .mcool | imargi-processing-pairs.cwl |
| Repli-seq[3] | Alignment | bwa-mem | .bam | repliseq-parta.cwl |
| | Filtering | samtools | - | |
| | Binning & aggregation | bedtools | .bw, .bg | |
| CUT&RUN[4] | Alignment & filtering | bowtie2, Picard, samtools, bedtools | .bam, .bedpe | cut-and-run-processing.cwl |
| | Peak calling | SEACR | .bw, .bg, .bed | cut-and-run-postaln.cwl |
| ATAC-seq[5] | Alignment & filtering | bowtie2, bedtools | .bed | atac.wdl |
| | Peak calling | MACS2 | .bw, .bigbed | |
| ChIP-seq[6] | Alignment & filtering | bwa, bedtools | .bed | chip.wdl |
| | Peak calling | MACS2, SPP | .bw, .bigbed | |
| RNA-seq[7] | Alignment | STAR | .bam | rna-seq-pipeline.wdl |
| | Expression quantification | RSEM | .tsv | |
| | Read coverage | STAR | .bw | |

Listed below are (i) subdirectories for Docker images from https://hub.docker.com/r/4dndcic; (ii) subdirectories from github repositories at https://github.com/4dn-dcic/ that hold the CWL or WDL pipeline files; and (iii) subdirectories for more information from https://data.4dnucleome.org/resources/data-analysis/. Note that for the Repliseq pipeline as well as the WDL pipelines from ENCODE, there is only one workflow file for the whole pipeline.
[1]4dn-hic, docker-4dn-hic/tree/v43/cwl, hi_c-processing-pipeline.
[2]imargi, iMARGI-Docker/tree/v1.1.1_dcic_4/src/cwl, imargi-pipeline.
[3]repliseq, docker-4dn-repliseq/tree/v16/cwl, repli-seq-processing-pipeline.
[4]cut-and-run-pipeline, docker-4dn-cut-and-run-pipeline/tree/v1/cwl, cut-and-run-pipeline.
[5]encode-atacseq, atac-seq-pipeline, atacseq-processing-pipeline.
[6]encode-chipseq, chip-seq-pipeline2, chipseq-processing-pipeline.
[7]encode-rnaseq, rna-seq-pipeline, rnaseq-processing-pipeline information.

visualized as a 1D genome track in HiGlass, and a corresponding quality control report. The peak calling step of these processing pipelines also yields optimal peaks and conservative peaks, which are both available on the portal in bigBed format. For RNA-seq, the final output is a bigWig file that contains read counts, and can be visualized as a 1D track in HiGlass. In addition, .tsv files of gene expression and isoform expression are also available. More information is available on the portal at the following links: https://data.4dnucleome.org/resources/data-analysis/chipseq-processing-pipeline for ChIP-seq; https://data.4dnucleome.org/resources/data-analysis/atacseq-processing-pipeline for ATAC-seq; and https://data.4dnucleome.org/resources/data-analysis/rnaseq-processing-pipeline for RNA-seq.

Several types of quality control and assessment are performed on submitted and pipeline generated results. FastQC[25] is run on all fastq files, and the report generated is stored in the cloud and viewable to users. PairsQC[26] is a software package developed to assess the quality of .pairs files generated by the Hi-C and MARGI pipelines, and these results are also available on the portal for viewing. Additional QC reports are available on the portal for .bam alignments, as well as ChIP-seq, RNA-seq, ATAC-seq, and Repliseq results, and report various metrics such as the percentage of reads mapped in the case of alignments, or overlap reproducibility measures in the case of ATAC-seq and ChIP-seq.

A primary advantage of using the 4DN Hi-C data is that all similar experiments have been processed in an automated fashion with the same software and versions, so the results are directly comparable and more amenable to meta-analysis. All data are processed by open source software from start (fastq files) to finish (contact matrices and domain calls). All intermediate files and full provenance graphs are available so that the user can easily find the processing steps and reproduce any portion of the full pipeline. All 4DN pipelines are available to download as Docker containers on Docker Hub (Table 2). On the AWS cloud, data processing pipelines employ Tibanna[42], developed by the 4DN DCIC. Tibanna queries metadata from the 4DN portal to obtain parameters for running the pipelines in the cloud, and updates metadata on the portal upon completion of pipelines to provide access to the processed results, which are stored in Amazon S3. For some of the other assay types, automated cloud pipelines are not yet available; in these cases, data submitters have the option to submit their own processed files that are stored and available in the cloud.

**Microscopy datasets**. Given the exciting developments in imaging technologies and their importance in investigating chromosomal dynamics, a considerable part of 4DN efforts are in the development and application of such techniques. Accordingly, the 4DN Data Portal was built to also handle microscopy datasets. Complementary to sequencing methods that measure average contact frequencies over large cell populations, imaging-based techniques such as FISH allow direct measurement of 3D distances between DNA loci in single cells. In addition to their usefulness in cross-validating findings obtained with high-throughput sequencing, microscopy data also provide valuable insight into cell-to-cell variability and dynamics that are lost in experiments with bulk samples.

The 4DN data portal hosts several types of microscopy experiments. To mention a few examples, ChromEMT combines an evolution of electron microscopy tomography with a novel DNA-labeling method, allowing visualization of chromatin organization in interphase and mitotic cells in situ at unprecedented resolution[43]. The OptoDroplet assay[44] can assess the condensation potential of proteins known to interact with membrane-less organelles in the nucleus[45]. Multiplexed FISH was used to obtain localization data of 8 Mb of human chromosome 19[7]. Single-particle tracking (SPT) experiments to

study nuclear protein dynamics[46] are also present on the portal. High-throughput FISH represents one example of integrating imaging and sequencing. This technique allows the examination of heterogeneity in genome organization by systematically determining the spatial position and distances between combinations of genomic interaction pairs identified by Hi-C[47].

Together, the 4DN data portal hosts >600 microscopy experiment sets (Table 3) at the time of writing, divided into 27 datasets; a table is available at https://data.4dnucleome.org/microscopy-data-overview. Most of the data are provided as processed files: these are the most reusable and data-rich files and include, for example, locus pair distances from FISH experiments, localization files from high-throughput Single-Molecule Switching Nanoscopy[48], or Spatio-temporal trajectory coordinates from SPT assays. A selection of raw image files is also available on the 4DN data portal and can be explored directly or downloaded for independent analysis. Microscopy files, unlike sequencing data, do not undergo any automated processing or quality control measures on the data portal at this time.

**Biosamples and tiered cell lines**. Data hosted at the 4DN data portal come from a variety of biological sources. These sources are mostly human and mouse, but fly, zebrafish, chicken, hamster, and green monkey experiments can also be found. The majority of experiments were performed on cell lines, but a few are derived from tissue samples or whole organism samples. Some biosamples derive from sources such as HeLa cells, for which access needs to be restricted. For these data, we make processed results available on the data portal, but raw files are not available for download. Instead, a dbGaP identifier is linked, and the user can request access from dbGaP if necessary.

Cell lines at the 4DN data portal can be categorized into three "tiers": Tier 1, Tier 2, and untiered. Early in the consortium, 4DN Network members had decided on a group of cell lines on which to perform coordinated experiments. The goal of this effort was to deliver data that are more directly comparable across different assays, whenever the biological question being studied would allow it. When a biosample is designated as Tier 1 in the 4DN data portal, it means that it is derived from an aliquot of cells obtained from a single common provider, to minimize sample-to-sample variation. Five human cell lines are designated as Tier 1: H1-ESC, GM12878, IMR90, HFF-hTERT (clone 6), and WTC-11. 11 different human and mouse cell lines are currently designated as Tier 2, indicating that multiple 4DN Network investigators have agreed to generate data on them. The untiered cell lines include any cell lines outside of the 4DN-designated lines. A complete list of the 4DN tiered cell lines can be found at https://4dnucleome.org/cell-lines.html.

**Table 3 Microscopy assay types in the 4D Nucleome Data Portal.**

| Experiment type | No. of public experiment sets |
| --- | --- |
| DNA FISH | 275 |
| Immunofluorescence | 138 |
| SPT | 101 |
| RNA FISH | 77 |
| Optodroplet | 13 |
| Electron Tomography | 3 |
| Multiplexed FISH | 1 |
| Total | 608 |

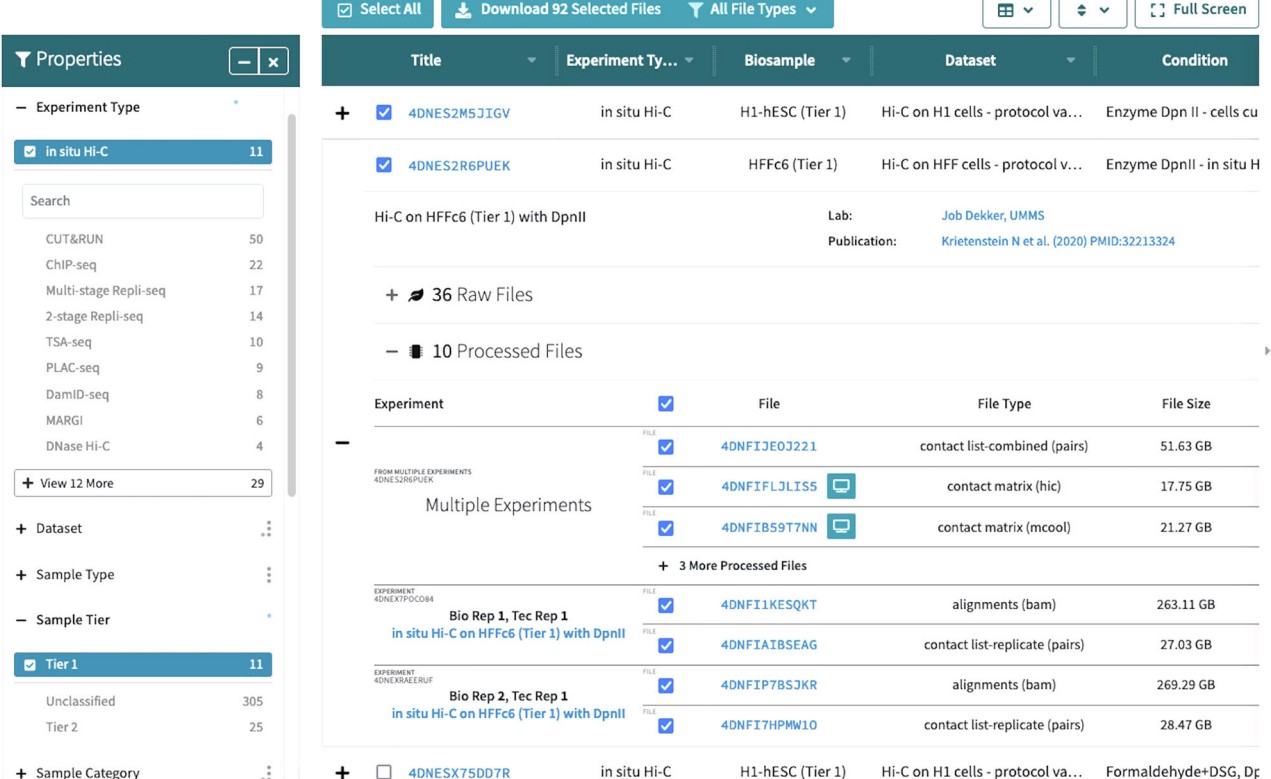

**Fig. 1 Browsing 4DN experiment sets.** The browse view features a table of experiment sets, the second of which can be seen expanded here to show additional metadata and information about files. On the left are a number of properties that can be used to filter the results; here "in situ Hi-C" is selected as well as Tier 1 samples. The top two experiment sets in the table are also shown with their checkboxes checked so that the "Download Files" button above the table can be used.

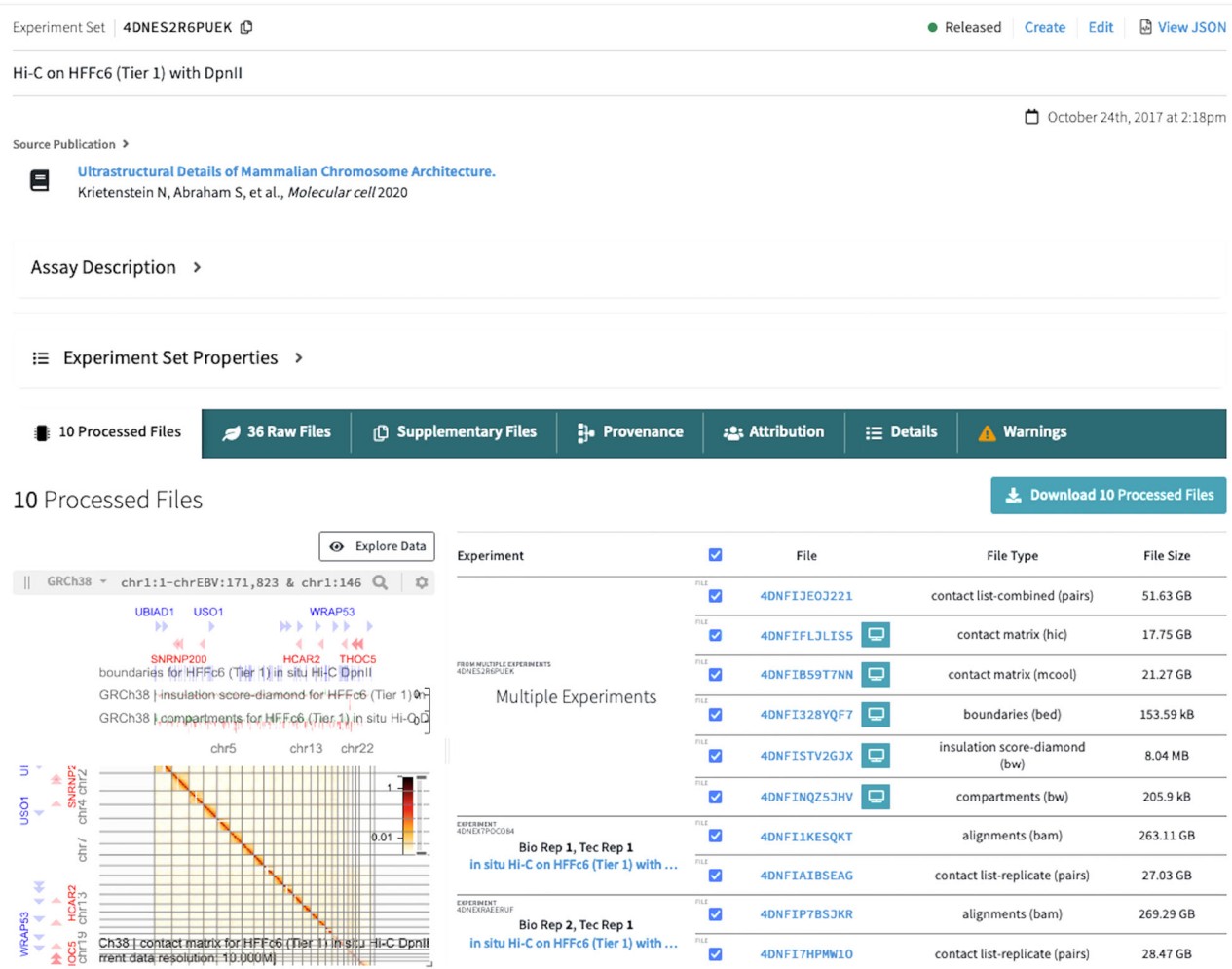

**Fig. 2 Item page for a replicate set of Hi-C experiments.** A source publication is shown near the top of the page, when available. Below these are two dropdown boxes: the first, titled Assay Description, can be expanded for an explanation of the assay, and the second, titled Experiment Set Properties, contains a selection of basic metadata fields. Below is a window with several tabs. The selected tab shows the processed files associated with the experiment. The .mcool contact matrix file is visualized on the left using the integrated HiGlass browser as a 2D track, with TAD boundaries, insulation scores, and compartments as 1D tracks above it. This display can be expanded for further data exploration. Scrolling down on this page would reveal the quality control metrics associated with the processed data.

**Data Portal services**. In the following sections, we describe some of the major functionalities of the 4DN data portal. A key aspect of a useful data portal is to give an overview of the type and amount of data available and allow the user to quickly find the datasets of interest. In addition to uniformly processed raw data, extensive metadata is collected and curated, enabling searches on various aspects of the data.

**Browsing datasets**. Finding datasets can be done either by browsing or by searching. To search, one can just click the search bubble in the top navigation bar, and a bigger search box will appear. Here the user can enter search terms, and optionally choose whether they want to search all items, or search by accession or within a particular item type. To browse, the 4DN data portal homepage (https://data.4dnucleome.org/) has a button called "Data" in the navigation bar with the following options: Browse All, Browse Sequencing, View Microscopy, and Browse by Publication. Clicking on "Browse All" will load a page of all public Experiment Sets. On the left side of the page is a number of filtering options, where experiments can be filtered on various details. In the example shown (Fig. 1), selecting filters of "in situ Hi-C" under experiment type and "Tier 1" under Sample Tier filters results in a smaller number. The columns labeled "Dataset" and "Condition" in the result table are useful in finding datasets of interest. Experiment Sets with the same "Dataset" name generally were generated together as part of a single study or analysis; the "Condition" field explains the differences between them. This is particularly useful when one set of replicates represents a control while another represents a treatment. Together the "dataset" and "condition" fields allow users to easily see which experiment sets go together, and what the experimental differences are.

Each result in the table can be expanded with the "+" button. The expanded view provides access to more details such as available files and the data-generating lab. Files from multiple experiment sets can be downloaded in bulk in the browse view here. Downloads require an account and a corresponding access key, but accounts can be created immediately and are free and accessible to anyone.

**Item pages: genomics**. Clicking on one of the accessions found in the browse results table will bring the user to that experiment set's item page (example shown in Fig. 2). More extensive metadata is

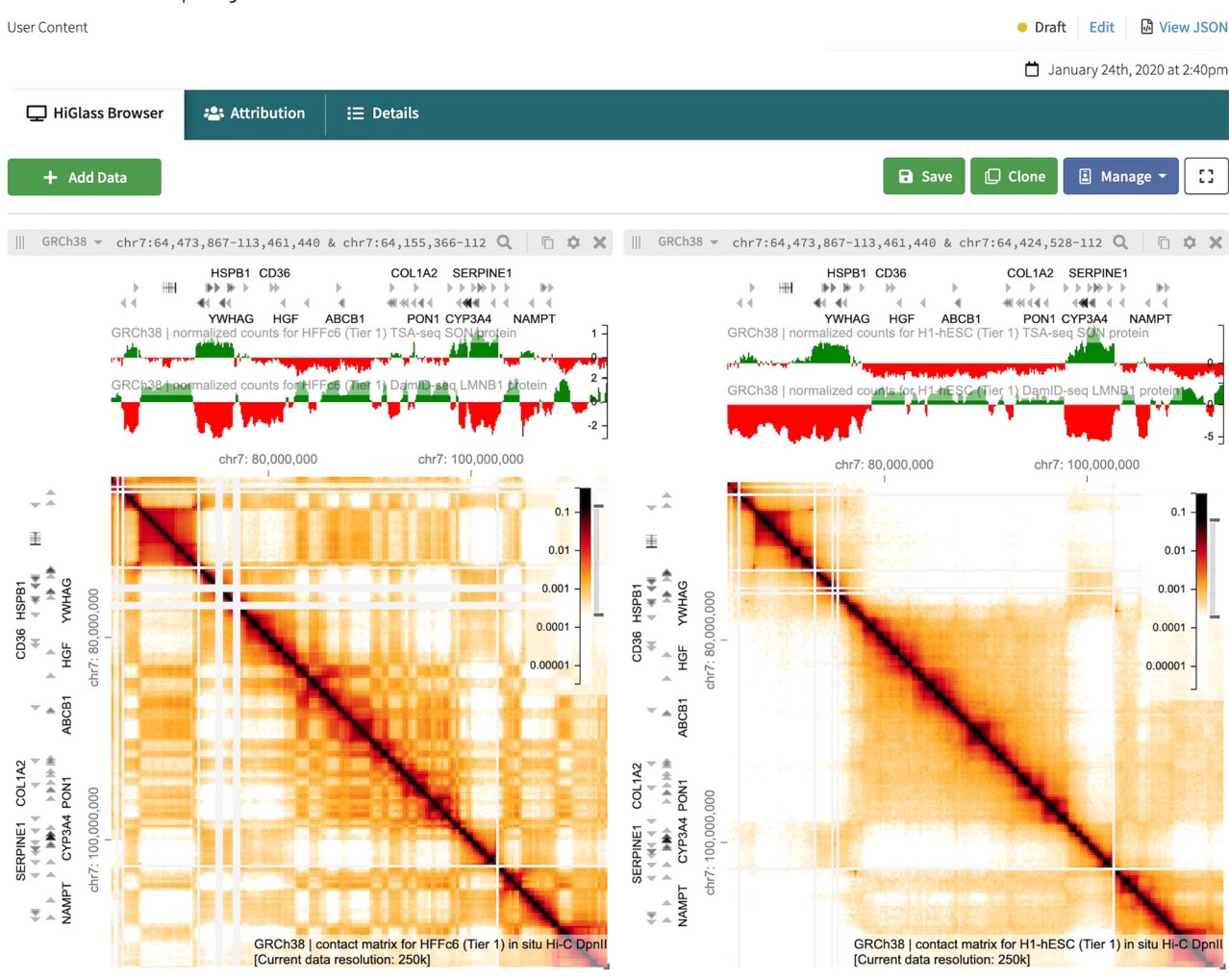

**Fig. 3 HiGlass Display containing 2D and 1D Genomic Tracks.** The display shows a visualization of an HFF in situ Hi-C contact matrix on the left, and one for H1 hESCs on the right. Above each 2D contact matrix visualization are 1D tracks from a TSA-seq experiment (topmost) and a DamID-seq experiment (second from top), in the cell type that matches the corresponding matrix. The "Add Data" button in the top left gives the user the option to add more files to the visualization display; on the top right, the user has the ability to save the display, clone the display to create a new item, or to manage permissions of who can view the display.

available on these pages. A reference publication will be featured at the top if the dataset has been published. If the dataset has not been published, it is still available for use, with the following data usage guidelines: (1) the data-generating lab should be contacted to discuss possible coordinated publication; (2) the 4DN white paper[1] should be cited; and (3) the lab which generated the data should be acknowledged. Further below on the page is a pane with several tabs, each providing more details about a different aspect of the experiment set. Typically these include Processed Files, Raw Files, Provenance, Attributions, Details, and Warnings or Commendations. The Raw Files and Processed Files tabs provide metadata about files associated with the experiment set and also designate the replicate from which they were generated. For processed files, this tab often includes a small HiGlass view of the final processed output, which can be expanded into a new window. If processed files were generated by one of the 4DN processing pipelines, there will also be a provenance graph available to view in a subsequent tab. This directed graph shows the inputs and outputs for each step in the pipeline that was run on the experiment set. In the case of user-submitted processed files where a 4DN processing pipeline

was not run, a provenance graph will still be present, but there will only be a single step titled "File Provenance Tracking Workflow" and it will show only relationships between input and output files, as the portal does not have any metadata about details of the processing performed by the user.

The final tab on item pages for Experiment Sets, Experiments, and Biosamples will be Warnings or Commendations. Presently these are used to indicate if an experiment lacks replicates, or to indicate whether the biosamples have all of the requisite metadata. The biosamples with a gold commendation have extensive metadata information as well as a morphology image taken before harvesting. This allows people who want to reuse the data to know more about the cultures and whether they had been growing with typical morphology. If a sample does not meet one or more metadata requirements for the gold commendation, this will show up under Warnings with a message indicating which piece(s) of metadata is missing.

All item types will also have Attributions and Details tabs. The attribution tab indicates which lab(s) generated the data and any associated publications. Datasets published in journals often must

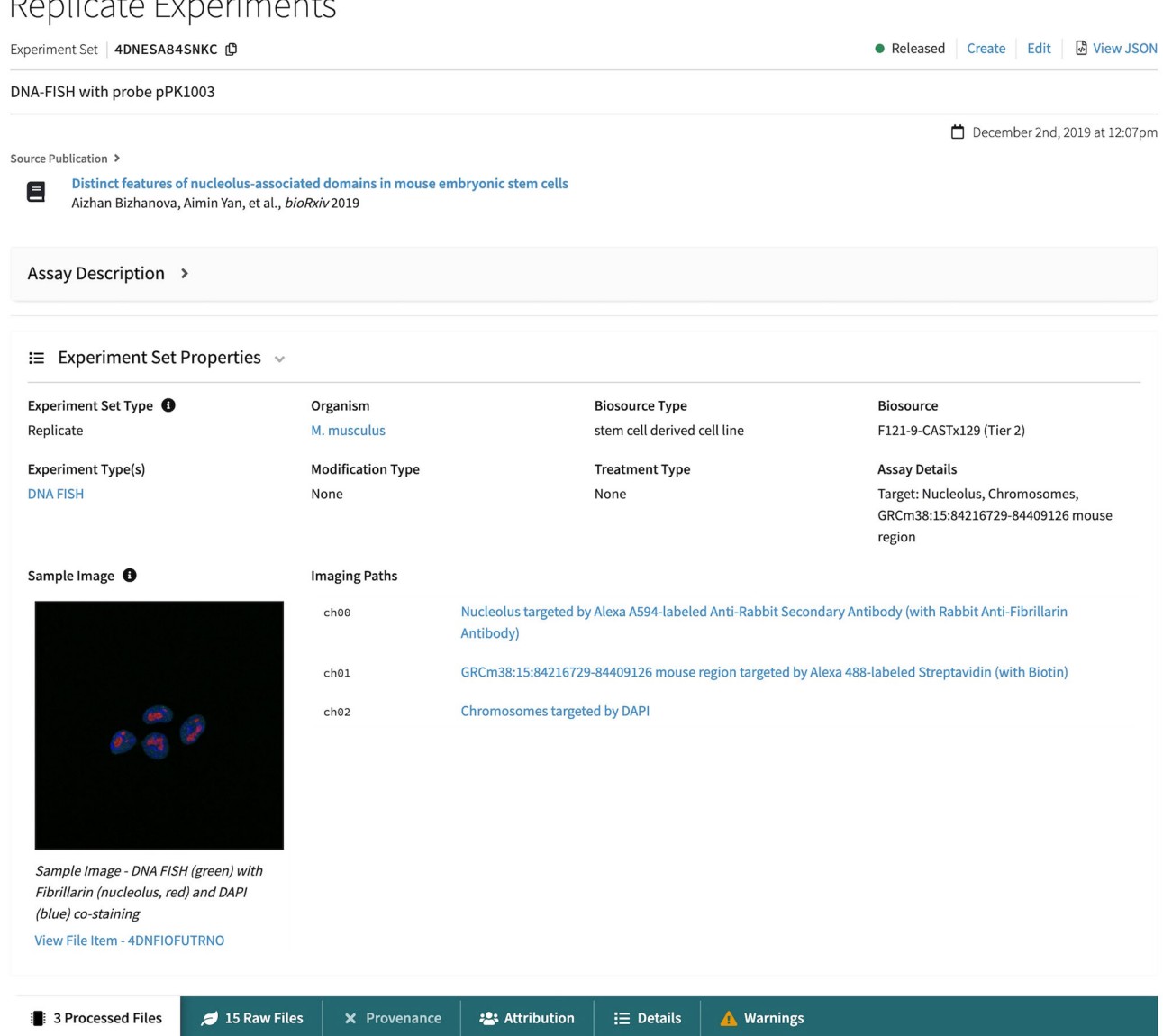

**Fig. 4 Microscopy experiment set item page.** Item pages for microscopy experiment sets are similar to those for genomics, but with a few important differences. The sample image field contains an image preview, which can be clicked for a popup containing an interactive image viewer. Beside the sample image is a field called "Imaging Paths", which describes what is imaged in each channel for the experiments, including the biological target and the antibodies or probes used.

also be deposited in GEO or SRA, and some 4DN datasets have also been deposited to the ENCODE portal; external datasets were often originally deposited at GEO and SRA before being added to 4DN. If available, identifiers to other databases that may also host the data are also included here. The details tab has a list of all metadata fields associated with the item, regardless of whether or not it is already displayed elsewhere on the page.

**Comparing files in HiGlass.** Whereas several genomic and epi-genomic data repositories have a 1D genome browser tool, con-tact frequencies across distant genomic locations require 2D visualization. To this end, a HiGlass browser[8] has been integrated into the portal for interactive visualization of both 2D contact matrices and 1D genomic tracks. The 4DN visualization work-space can be accessed either at https://data.4dnucleome.org/tools/visualization or by clicking on the "Explore Data" button above the miniature HiGlass view displayed on an Experiment Set item

page. A short tutorial video can also be found at https://www.youtube.com/watch?v=LEDaOa3NZtM.

The close integration of HiGlass into the portal allows users to leverage the portal search capabilities to identify files of interest for visualization that can then be added to existing HiGlass views. Files can be added to the HiGlass display by clicking on the Add Data button, which opens a window where users can filter via various metadata fields to find a file of interest. Multiple 2D contact matrices can be compared at once (Fig. 3), and the position and zoom level of each view are locked to the first one, such that zooming in or out or dragging the mouse to different genomic locations moves all the matrices in unison.

1D tracks can also be added to each HiGlass view. By default, each 1D track added will be added to every view in the display. This allows it to be compared to each 2D matrix simultaneously. Multiple 1D tracks can be added in this fashion. Users can then save their customized views for later retrieval and sharing with collaborators.

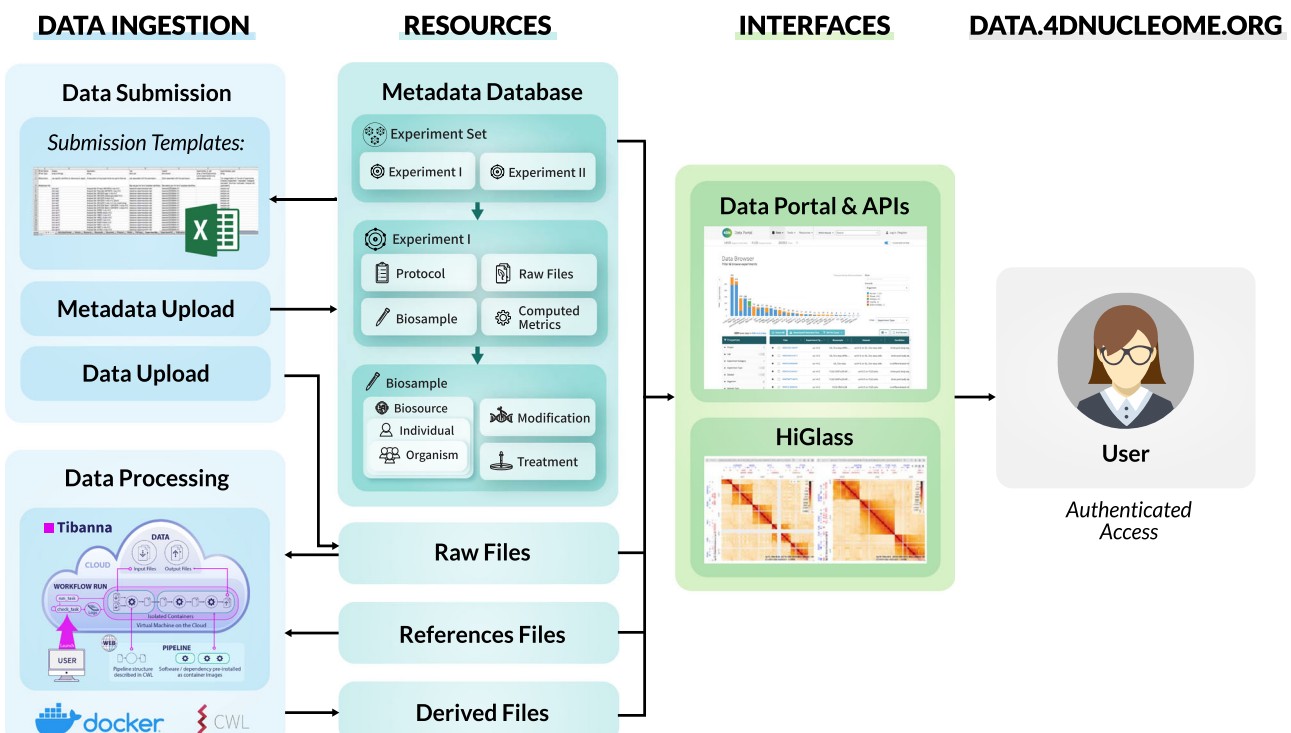

**Fig. 5 4DN Data Portal architecture.** Metadata is submitted by users via spreadsheet forms and gets loaded into the database, while associated files are uploaded into cloud storage. Once metadata and files are in place, automated processing pipelines can be run on AWS using the Tibanna pipeline runner. All metadata can then be searched via the data portal website, where files can also be visualized with HiGlass. The data portal website is accessible to external users, who can also login in order to download files.

**Finding microscopy experiments**. In the Data menu in the top navigation bar, clicking on "View Microscopy Datasets" will bring the user to a table of the microscopy datasets that can be found in the portal. These are organized by Dataset label rather than by Experiment Set, as some datasets comprise large numbers of replicate sets, allowing the table to be much more concise and straightforward. The second column contains a link to a browse page of all the Experiment Sets in the dataset.

A microscopy experiment set item page (Fig. 4) is organized much the same as the sequencing experiment sets, with a few notable differences. In the overview section above the tabs, in addition to common metadata, there may be two extra fields: a sample image, for sets that include raw files, and information about the imaging paths. The Imaging Path refers to metadata about what was imaged, including cellular target, antibodies or probes, and the detected label. In addition, there are no processing pipelines run by the 4DN data portal on microscopy datasets, so the Provenance tab will often be dimmed indicating no provenance is available. However, in some cases submitting laboratories indicate which files are processed results derived from others and in these cases, a simple provenance graph reflecting this is shown.

If a sample image is present and derives from a raw image file on the portal rather than a rendered jpg, then clicking on the image will bring the user to an interactive image viewer, where it is possible to scroll through focal planes in a z-stack, or through time points in a time-lapse image, or adjust signal and color levels in the display.

## Discussion

With 1273 sequencing and 608 microscopy experiment sets that are publicly available and consistently curated, the portal provides a high volume of data in a wide array of data types to nuclear architecture researchers across the globe. The portal interface provides user-friendly search capabilities with many options for filtering results, and the result pages have been designed to communicate extensive metadata for each item. Uniform processing on assays with automated pipelines also ensures that the processed results are consistent and directly comparable.

The integrated HiGlass browser allows the data portal to serve not only as a hub for searching and downloading research data but also as a visualization platform. 2D output from genomic assays like Hi-C can be directly compared to other 2D assays like DNA SPRITE, or to results from 1D assays like ChIP-seq or CUT&RUN. Even when only looking at Hi-C experiments, some 4DN data contributors have uploaded experiments comparing different variations in Hi-C protocols[11], and the HiGlass browser provides an easy way to examine differences in data output resulting from protocol modifications.

Prior to the development of the 4DN data portal, there was a lack of a centralized repository that specializes in chromatin conformation and nuclear structure data, specifically. Other repositories that host Hi-C and similar datasets include the ENCODE portal[49], as well as NCBI's GEO[50] and EMBL-EBI's ArrayExpress[51]. GEO and ArrayExpress are great resources for hosting published sequencing datasets, but they lack a finer specialized focus or visualization capabilities for processed results. Their metadata also tends to be minimal and less structured, which is understandable given the huge diversity of datasets that need to be hosted there. ENCODE has an infrastructure that is very similar to the 4DN data portal, and it has amassed an impressive number of sequencing datasets over the years it has been active. Although they have Hi-C datasets available, since their portal has been engineered to focus more on 1D datasets, they lack a 2D genome browser and the Hi-C datasets cannot be visualized there. The 4DN data portal thus fills a very important niche: it represents a centralized repository for nuclear structure

data specifically, with a data model tailored to provide extensive metadata about each experiment; it enables visualization of processed results, where 2D genomic data can be compared to 1D genomic tracks; and it also houses microscopy data that can complement sequencing results. To our knowledge, this is unique to the 4DN data portal.

The second phase of the NIH Common Fund 4D Nucleome Project began in 2020, and will proceed for five years, with a focus on the role of nuclear structure and function in human health and disease, as well as on the continued development of data visualization and integration tools. Thus, we expect the data volume in the 4DN data portal to continue to increase. Development on the portal is ongoing to ensure that the 4DN data portal continues to serve the needs of the nuclear biology research community.

## Methods

**Portal architecture**. The 4DN Consortium was projected to generate large volumes of diverse datasets at its initiation. To be useful to scientists with different technical and scientific backgrounds, the primary objectives were to ensure users can determine what data are available on the portal, and that data of different modalities should be made accessible to all scientists. This requires an architecture that is modular, responsive, and scalable, with an easily extensible data model. Our software architecture is entirely cloud-based, as we envisioned that taking analytical tools to the cloud environment where data reside, rather than downloading large amounts of data to a local server, is quickly becoming the preferred mode of data analysis.

The design of the 4DN data portal infrastructure (Fig. 5), originally based on the ENCODE infrastructure, includes the following components: (1) A postgres database storing metadata in json format, first developed in ENCODE; (2) The python pyramid framework for the database known as SnoVault[52], first developed in ENCODE but further tailored and developed by the 4DN DCIC (https://github.com/4dn-dcic/snovault); (3) The FourFront front-end (https://github.com/4dn-dcic/fourfront), originally based on EncodeD[52] from ENCODE, but engineered by 4DN DCIC to feature a data model for representing diverse datasets, and includes a modern front-end with reactJS to provide a responsive user experience; (4) Elasticsearch that provides fast and efficient search with various metadata fields by indexing all items and formatting them for retrieval; (5) AWS S3 used for file storage, enabling all public data files to be accessed via the data portal interface; (6) A RESTful API underlying the infrastructure, through which all metadata in the portal can be accessed.

In the AWS cloud environment, changes in the database are indexed within seconds to minutes, allowing updates and releases of datasets with minimal overhead. Similarly, the infrastructure was developed with features that allow the release of changes to a data model and software versions overnight without any server downtime. Thus, we are able to ingest and release datasets with new data models at any time, without having to enforce data release cycles.

All of the software used in the data portal is open access and open source (https://github.com/4dn-dcic/). Finally, the 4DN data portal also has an associated support contact email (support@4dnucleome.org), allowing curators to quickly address any question or concern from end-users.

**Reporting summary**. Further information on research design is available in the Nature Research Reporting Summary linked to this article.

## Data availability

All datasets described are available at https://data.4dnucleome.org/.

## Code availability

All of the software that comprises the 4DN data portal infrastructure is free and open source. All code repositories mentioned are available from https://github.com/4dn-dcic.

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

## Acknowledgements

We would like to thank the current and past members of the 4DN Network for all the time and effort they put into data generation, and without whom this data portal would not have been built. We also thank J. Michael Cherry and the development team at ENCODE for helpful discussions and advice regarding setting up and running a data portal, as well as for allowing us to build off of the ENCODE software ecosystem. This work was funded by the NIH Common Fund grant 1U01CA200059 to P.J.P.

## Author contributions

S.B.R. wrote the manuscript with the help of A.J.S., A.C., K.K., C.B., L.M., B.H.A., and P.J.P. J.J., A.K.B., C.V., S.B.R., A.J.S., K.K., A.C., C.B., L.M., S.L., A.D.V., W.R., K.M.P., P.K., N.A., M.I., S.U.O., and U.C. developed the software. A.J.S., K.K., A.C., S.B.R., and L.M. handled data submissions. S.R.E. carried out graphic design. P.J.P. supervised the work along with N.G., L.A.M., and B.H.A.

## Competing interests

The authors declare no conflict of interest.
