## [Peer Review File · Nature Communications]

REVIEWER COMMENTS

Reviewer #1 (Remarks to the Author):

The authors present a manuscript I which they describe access to their data portal which stores the data generated by the groups in the 4D Nucleome consortium. The paper gives an elegant overview of the stored data and how it can be accessed. I have only minor comments.

Minor points:

1. Please provide details on specific cooltools parameters used for compartments and TAD calling procedure in the main text.
2. In the current version of the manuscript the description of ChIP-seq, RNA-seq and ATAC-seq is somewhat short. The text just states "For ATAC-seq, ChIP-seq, and RNA-seq data, pipelines from the ENCODE Data Coordination Center have been adapted for the 4DN platform" and requires extra search for this information in the external sources. Please consider including links to the technical details description from the 4DN data portal. (<https://data.4dnucleome.org/resources/data-analysis/chipseq-processing-pipeline>, <https://data.4dnucleome.org/resources/data-analysis/atacseq-processing-pipeline>, <https://data.4dnucleome.org/resources/data-analysis/rnaseq-processing-pipeline>).
3. A table with description of microscopy datasets available at 4DN data portal (similar to Table 1 for genomic assays) would give a better overview of them and would benefit the reader.
4. Is there any walkthrough for the HiGlass browser? If not, please consider incorporating one into the 4DN data portal. It may benefit the researchers who are not familiar with this tool and facilitate their use of genome structure visualizations.
5. The search button location on the main page of the 4DN data portal is counter-intuitive and invisible. The reviewer urges authors to make it more clear, like on the ENCODE data portal (<https://www.encodeproject.org/>, Search ENCODE field). If the user doesn't notice the search field and has to look up a particular dataset by selecting multiple filtering options, the interface becomes very user-unfriendly.
6. Does the 4DN Network have any plans to increase the number of available External datasets? Storing data from the community and some common datasets is a good strategy for now. However, there is a lot of data available through NCBI GEO which is equally important, and the community can benefit from the use of uniformly processed data. Some suggestions regarding this issue:
 - Add a form for authors if they want to submit their data to be processed and stored in 4DN data portal
 - Ask 3D genome community which datasets they would like to be present in 4DN data portal

Reviewer #2 (Remarks to the Author):

The manuscript titled "The 4D Nucleome Data Portal: a resource for searching and visualizing curated nucleomics data" by Reiff et al described an online data repository portal specialized for chromatin conformation and nuclear structure data. The manuscript is

well written clearly describing the mission of the data portal, its data content and how the data were generated and uniformly processed, and how users can search and visualize data of their interest through the intuitive web interface. The 4D Nucleome Data Portal should be a very useful and unique resource to the nuclear biology research community.

The reviewer has the following comments and minor suggestions about possible improvements of the data portal:

1. The manuscript does not mention data access and usage policies, particularly for human data. Is there a data access control committee to grant access for controlled human data, or it's covered by dbGaP?

2. The portal contains externally generated data from publications, here is an example: <https://data.4dnucleome.org/files-fastq/4DNFI533DTK2/#file-overview>. It's unclear whether the raw data was uploaded by the original data producer or downloaded from another data repository. If it's latter, it would be helpful to include a reference on the portal's file page to the corresponding data entity of the other repository.

3. While exploring the data portal, the reviewer found the Search View is very useful: <https://data.4dnucleome.org/search/?type=Item>, particularly when looking for specific type of items, such as a particular type of data file or workflow, which are not possible through the Browse All page that is centred around Experiment Sets. It should be helpful to provide an entry point directly to the Search View, maybe under the Data menu.

4. Table 2 includes information of analysis pipelines with underlying software tools and docker images, which are helpful. However, in order to allow others to reproduce the analysis, it's necessary to include the workflow code of these pipelines. Such information seems available on the portal, for example this page: <https://data.4dnucleome.org/workflows/4DNWF84NFOQP/#details> provides CWL directory URL and CWL main file name. Can this information be added to Table 2 for all pipelines?

5. Table 2, footnote c include unnecessary docker hub url, it should be modified to match all other footnotes.

6. It's unclear what File Provenance Tracking Workflow is, like on this page: <https://data.4dnucleome.org/files-processed/4DNFI82S5W3G/#graph-section>, was the mcool file produced by this File Provenance Tracking Workflow? What are the actual computational analysis steps? The other mcool file here: <https://data.4dnucleome.org/files-processed/4DNFI96GX25S/#graph-section> seems contain much more reasonable provenance information on analysis steps. What's the difference here?

Response to Reviewers

We have done our best to address the reviewers' comments in our revision (and in the data portal, as appropriate). To summarize, we have added a new table that details the types and numbers of microscopy assays in the data portal; we have made the search box more obvious in the portal's navigation bar; we have added CWL filenames and links in Table 2 describing the analysis pipeline workflows; and we have added links and clarifying text in the manuscript where appropriate. In addition to changes suggested by the reviewers, we have also changed the numbers of experiment sets to reflect the most up-to-date totals, as we have released more datasets since our initial submission. Finally, we have adjusted the section headings to be consistent with the formatting instructions, and moved the subsection on "Portal Architecture" to a Methods section at the end of the manuscript.

We provide point-by-point responses to the reviewers' comments below. Our comments are in black while the reviewer comments are indented and in grey; any texts changed in the manuscript are indicated in red in the revised pdf.

Responses to Individual Reviewer Comments

Reviewer #1 (Remarks to the Author):

1. Please provide details on specific cooltools parameters used for compartments and TAD calling procedure in the main text.

We have added this to the main text, in the paragraph where we describe compartment calling and TAD calling (page 4 of the revised pdf, lines 222-229 and 233-238).

2. In the current version of the manuscript the description of ChIP-seq, RNA-seq and ATAC-seq is somewhat short. The text just states "For ATAC-seq, ChIP-seq, and RNA-seq data, pipelines from the ENCODE Data Coordination Center have been adapted for the 4DN platform" and requires extra search for this information in the external sources. Please consider including links to the technical details description from the 4DN data portal.

<https://data.4dnucleome.org/resources/data-analysis/chipseq-processing-pipeline>,
<https://data.4dnucleome.org/resources/data-analysis/atacseq-processing-pipeline>,
<https://data.4dnucleome.org/resources/data-analysis/rnaseq-processing-pipeline>).

We have added these links to the paragraph about the pipelines adapted from ENCODE (page 5 of the revised pdf, lines 284-291). We are close to the limit on the word count, and we had to keep some of our descriptions brief.

3. A table with description of microscopy datasets available at 4DN data portal (similar to Table 1 for genomic assays) would give a better overview of them and would benefit the reader.

We've now added an additional table, Table 3, which shows the different microscopy experiment types in the 4DN data portal and how many public experiment sets we have of each (page 6). Since the entire table is new, we did not highlight the whole table to indicate changes, but a reference to Table 3 was added to the main text and is indicated in red.

4. Is there any walkthrough for the HiGlass browser? If not, please consider incorporating one into the 4DN data portal. It may benefit the researchers who are not familiar with this tool and facilitate their use of genome structure visualizations.

In response to this suggestion, we decided to record a 5 minute HiGlass tutorial video that explains how to use HiGlass visualization in the 4DN data portal (available on the 4DNucleome YouTube channel at <https://www.youtube.com/watch?v=LEDaOa3NZtM>). We have also linked this on our Visualization page at <https://data.4dnucleome.org/tools/visualization>, and in our FAQ at https://data.4dnucleome.org/help/user-guide/faq#data_exploration. We have also mentioned this video briefly in the manuscript, in the section titled "Comparing Files in HiGlass" (page 9, lines 523-524).

5. The search button location on the main page of the 4DN data portal is counter-intuitive and invisible. The reviewer urges authors to make it more clear, like on the ENCODE data portal (<https://www.encodeproject.org/>, Search ENCODE field). If the user doesn't notice the search field and has to look up a particular dataset by selecting multiple filtering options, the interface becomes very user-unfriendly.

We have kept the search button in the top navigation bar, but we made it more obvious by adding a box around it and "Search ..." text inside. Clicking on it now expands the box and allows the users multiple options for searching, including All Items, By Accession, Experiment Sets, Publications, Files, and Biosources. We have also added a couple sentences that describe this in the beginning of the "Browsing Datasets" section (page 7, lines 417-424).

6. Does the 4DN Network have any plans to increase the number of available External datasets? Storing data from the community and some common datasets is a good strategy for now. However, there is a lot of data available through NCBI GEO which is equally important, and the community can benefit from the use of uniformly processed data.

Some suggestions regarding this issue:

- Add a form for authors if they want to submit their data to be processed and stored in 4DN data portal
- Ask 3D genome community which datasets they would like to be present in 4DN data portal

We do have interest in providing more external datasets that are useful to the community, but of course our ability to do this is limited by curator time and needs to be balanced with other development priorities. Curating GEO data can be particularly time-consuming because the metadata requirements for GEO are less stringent than those for 4DN, ENCODE, and other NIH consortia, and we'd have to contact the submitters to sort out the details in many instances. In the past, there have also been

suggestions to collect entirely databases that have epigenetic data, e.g., Cistrome, but such projects are beyond the scope of our project. While we don't have a form, we have a helpdesk email (support@4dnucleome.org) that can be used to make requests or for researchers to inquire about data submission. We have added a sentence to the first paragraph in results (on page 2 of the revised pdf, lines 106-109) that makes this explicit. The FAQ in our help pages also has a question and answer about submitting external datasets where we provide our helpdesk email (<https://data.4dnucleome.org/help/user-guide/faq#can-i-submit-data-if-i-am-not-affiliated-with-4dn>).

Reviewer #2 (Remarks to the Author):

1. The manuscript does not mention data access and usage policies, particularly for human data. Is there a data access control committee to grant access for controlled human data, or it's covered by dbGaP?

Nearly all of the data on the portal are not subject to data access restrictions. For controlled access biosources, such as HeLa cells, we make the processed results available on the portal but the raw files are not available for download. We have added a couple sentences to the first paragraph under "Biosamples and Tiered Cell Lines" that address this (page 7, lines 383-388). Instead a dbGaP identifier is linked, and the user can request access from dbGaP if necessary.

2. The portal contains externally generated data from publications, here is an example: <https://data.4dnucleome.org/files-fastq/4DNFI533DTK2/#file-overview>. It's unclear whether the raw data was uploaded by the original data producer or downloaded from another data repository. If it's latter, it would be helpful to include a reference on the portal's file page to the corresponding data entity of the other repository.

On that particular page, if you click to the Attribution tab (<https://data.4dnucleome.org/files-fastq/4DNFI533DTK2/#attribution>) the SRA accession is listed. We mentioned the attribution tab and external references to other databases in the manuscript, in the last paragraph under "Item pages: Genomics", but we have added a little additional text that might clarify that dbxrefs for external datasets can also be found in the Attribution tab (page 8, lines 505-507).

3. While exploring the data portal, the reviewer found the Search View is very useful: <https://data.4dnucleome.org/search/?type=Item>, particularly when looking for specific type of items, such as a particular type of data file or workflow, which are not possible through the Browse All page that is centred around Experiment Sets. It should be helpful to provide an entry point directly to the Search View, maybe under the Data menu.

We had a search icon in the top navigation bar, but as the previous reviewer mentioned it was hard to see. We responded to that comment above, but to briefly reiterate, we made the search button more

obvious and clicking on it now allows the users more searching options in a dropdown menu (changes to the manuscript text are under “Browsing Datasets” on page 7, lines 417-424).

4. Table 2 includes information of analysis pipelines with underlying software tools and docker images, which are helpful. However, in order to allow others to reproduce the analysis, it's necessary to include the workflow code of these pipelines. Such information seems available on the portal, for example this page: <https://data.4dnucleome.org/workflows/4DNWF84NFOQP/#details> provides CWL directory URL and CWL main file name. Can this information be added to Table 2 for all pipelines?

We have added information to Table 2 (page 5) about the cwl files - the CWL or WDL filename for the step(s) are in the table, while the url is in the table footnotes. The WDL files for the ENCODE pipelines are also indicated, although for these there is one file per pipeline rather than for each step (and this is also true for the Repli-seq pipeline). Table 2 was reorganized and corrected slightly to be more accurate with respect to how the workflows in the pipeline are organized; for instance, CUT&RUN now has “alignment” and “filtering” in the same row because these steps are both in the CWL file indicated in the new last column. Additionally we also corrected an error in the main text and in the table, which stated that merging replicates was performed after matrix aggregation in the Hi-C pipeline; the corrected text states that merging replicates is done prior to matrix aggregation (top left of page 4, lines 177-179).

5. Table 2, footnote c include unnecessary docker hub url, it should be modified to match all other footnotes.

We have renamed the repliseq docker link text to be consistent with the other table footnotes (Table 2, page 5).

6. It's unclear what File Provenance Tracking Workflow is, like on this page: <https://data.4dnucleome.org/files-processed/4DNFI82S5W3G/#graph-section>, was the mcool file produced by this File Provenance Tracking Workflow? What are the actual computational analysis steps? The other mcool file here: <https://data.4dnucleome.org/files-processed/4DNFI96GX25S/#graph-section> seems contain much more reasonable provenance information on analysis steps. What's the difference here?

The difference is that the file with detailed provenance was processed by the 4DN data portal, but the file with only the “File Provenance Tracking Workflow” step was processed by the submitter and uploaded. We don't track metadata about external processing steps, as this is not reasonable under the current data model, so this graph is just meant to link which input files were used to generate the output, but the submitting lab can be contacted for details. This was mentioned briefly in the manuscript, in the first paragraph under “Item pages: Genomics”, but we have added some additional text there that explicitly includes the name of this workflow (page 8, lines 481-485). In addition, on each file page that represents user-submitted results, as in the link the reviewer added, we have now added a note at the top of the page that more explicitly states that these results are uploaded by the lab and are

not results from 4DN standardized pipelines, and thus do not have full metadata regarding workflow steps. This should be visible to you when clicking the link you provided for file 4DNFI82S5W3G.

Other miscellaneous revisions:

- Numbers of experiments in the portal have been updated in the manuscript wherever they appear.
- Tables have been modified according to the formatting instructions, and Table 1 has an added footnote indicating what the bold rows signify.
- Single-Molecule Switching Nanoscopy had been mistakenly referred to as “Small Molecule Switching Nanoscopy”, this was corrected (page 6, line 368 in the revised pdf).
- The subsection titled “Portal Architecture” has been moved to a new “Methods” section after the Discussion.

REVIEWERS' COMMENTS

Reviewer #1 (Remarks to the Author):

This reviewer appreciates the effort made by the authors to improve the manuscript and the 4D Nucleome Data Portal. The new Table 3 gives a good overview of the microscopy data available through the portal. The additional pipelines description in Table 2 and in the "Genomics Pipelines and QC" section reduce the time necessary to understand the data processing steps and make it easier to find the code. The reviewer would like to additionally acknowledge the recorded HiGlass walkthrough on YouTube that is definitely going to help future users of the portal. Lastly, it is now also clear that the 4DN encourages external scientists to freely submit their data to the portal. In summary, the authors have convincingly addressed all the concerns.

Reviewer #2 (Remarks to the Author):

In the revised manuscript, the authors have properly addressed all comments raised previously. The reviewer suggests publication.